# Causes of morbidity and mortality among patients admitted in a tertiary hospital in southern Nigeria: A 6 year evaluation

Henshaw Uchechi Okoroiwu[1]*, Kingsley Ikenna Uchendu[2], Rita A. Essien[3]

1 Haematology Unit, Department of Medical Laboratory Science, University of Calabar, Calabar, Nigeria, 2 Clinical Chemistry Division, Department of Medical Laboratory Science, University of Nigeria, Nsukka, Nigeria, 3 Department of Haematology and Blood Transfusion, University of Calabar Teaching Hospital, Calabar, Cross River State, Nigeria

* okoroiwuhenshaw@gmail.com

**Data Availability Statement:** All relevant data are within the manuscript and its Supporting Information files.

## Abstract

### Background

Data on morbidity and mortality are essential in assessing disease burden, monitoring and evaluation of health policies. The aim of this study is to describe the causes of morbidity and mortality in the wards of University of Calabar Teaching Hospital (UCTH).

### Methods

The study took a retrospective approach evaluating causes of morbidity and mortality from 2012–2017. Causes of death were documented based on International Classification of Disease 10 (ICD-10). Data were retrieved from health records department, UCTH.

### Results

Overall, 2,198 deaths were recorded out of the 49,287 admissions during the study period giving a mortality rate of 4.5% comprising 1,152 (52.4%) males and 1,046 (47.6%) females. A greater number of males were admitted via accident and emergency. Age group 15–45 years had the highest number of admissions (57.9%) and deaths (37.7%), while age group >65 years recorded the highest number of deaths per admission (9.7% mortality rate). The broad leading causes of death were infectious and parasitic disease and diseases of the circulatory system (cardiovascular diseases) accounting for 22.7% and 15.8% of all deaths, respectively. However, diseases of the circulatory system recorded the highest number of deaths per admission (13.7% mortality rate). Overall, infectious diseases were the chief cause of mortality in adults while conditions originating from perinatal period were the major cause of death in children. Septicemia (6.0%), stroke (4.2%), liver diseases (4.1%), tuberculosis (3.7%), diabetes (3.6%) and HIV/AIDS (3.4%) were the specific leading cases of deaths. Sepsis, chronic diseases of the tonsil and adenoids and malaria were the specific leading causes of death in children, while sepsis, stroke and liver diseases were the leading cause of death in adults.

**Funding:** The author(s) received no specific funding for this work.

**Competing interests:** The authors have declared that no competing interests exist.

**Abbreviations:** AIDS, Acqured Immune Deficiency Syndrome; HIV, Human immunodeficiency virus; ICD, International Classification of Diseases; MDR, Multidrug resistance; TB, Tuberculosis; UCTH, University of Calabar Teaching Hospital; WHO, World health organization.

## Conclusion

Most causes of deaths in this study are preventable. This study revealed double burden of communicable and non-communicable diseases.

## Background

Causes of morbidity and mortality are relevant parameters for documentation of the geographical burden of disease and for public health planning, involving programmatic needs, assessing intervention programmes, and reevaluation of health policies [1]. They are also relevant tools for keeping track of the health of populations as well as for effective response to changing epidemiological trends [2–4]. More so, they serve as tool for quality control of health care system. For instance, deaths that occur due to causes that should otherwise not be fatal at the instance of effective medical practice, known as amenable mortality is an indicator of national levels of personal health-care access and quality [2,5].

Globally, there were 56.9 million deaths in 2016 from varying causes among different regions [6]. Ischemic heart disease and stroke are the global leading causes of death, accounting for a combined 15.2 million deaths in 2016 and have remained the leading causes of death globally in the prior 15 years [6]. The global burden of disease study 2017 reported ischemic heart disease, neonatal disorders and stroke as leading causes of early death [7]. Greater proportion of developing countries have mortality pattern that show larger proportion of infectious disease and the risk of death during pregnancy and childbirth whereas cardiovascular diseases, chronic respiratory diseases and cancers account for most deaths in the developed world [8]. Population-based data on pattern of morbidity and mortality are often lacking in developing countries, hospital based pattern of morbidity and mortality often offer best alternative [4].

Most mortality reviews in Nigeria emanated from the south-west [9–13], north [14,15] and south east [16]. There is a paucity of data on morbidity and mortality pattern in the southern region. Hence, this study is aimed at bridging this gap by reviewing comprehensive data on morbidity and mortality in a tertiary institution in southern Nigeria.

## Methods

### Study design

This study took a retrospective descriptive cross-sectional method in analyzing causes of morbidity and mortality in University of Calabar Teaching Hospital (UCTH) from January 2012 to December, 2017.

### Study area

This study was conducted at University of Calabar Teaching Hospital, Calabar Cross River State, Nigeria, which is a 410 bed space capacity tertiary health care institution. The hospital is made up of 15 wards and 11 clinics. It is generally stratified into health care service department and administrative department, mortuary services and laundry and tailoring unit. The health care services is composed of laboratory department, nursing services, surgery, internal medicine, family medicine, pediatrics, obstetrics and gynecology, ophthalmology, physiotherapy, food and nutrition, orthopedics, accident and emergency, dental department, dialysis and blood bank units [17,18]. Though there is no published annual admission and mortality data,

there are more than 2000 deliveries annually in the centre [19]. Cross River State is one of the states that form the southern part of Nigeria with an area of 21,787km$^2$ and a population of 2,892,988 (using the 2006 census) [20,21]. The hospital is sited in Calabar metropolis which is a fusion of Calabar Municipality and Calabar South Local Government Areas (Fig 1) [22]

## Study population

Patients who were admitted or died (while in admission) within 2012 and 2017 were included in the study. More so, it is pertinent to note that female subjects admitted for labour and delivery were part of the study population. Patients brought in dead before arrival were excluded.

## Data collection

Data on demographics, causes of mortality and morbidity were retrospectively extracted from the health records department of the University of Calabar Teaching Hospital where they are coded based on international classification of disease– 10 (ICD—10) [23]. The various ICD classifications were allotted by the staff of health record department. The data are usually updated daily via outgone and returning patient case notes (folder) and are compiled into quarterly report submitted to Medical Advisory Committee. The data entry into spreadsheet from the original report book was performed with the assistance of 6 trained research assistants.

## Ethics approval

This study was approved by Health Research Ethical Committee (HREC) of the University of Calabar Teaching Hospital.

## Statistical analysis

Data generated in this study were entered and analyzed using SPSS version 22 (IBM Corps, Armonk, NY, USA). Frequencies and Percentages were used to represent the categorical variables. Pearson Chi square test was used to assess association between variables. Mantel Haenzel test of trend was used to assess linear association. Odd ratio was used to assess odd of occurrence in categorical variables. Alpha value was set at 0.05.

## Results

A total of 49,287 patients were admitted into the wards and accident and emergency (casualty ward) during the period of study. Of these, 2,198 died giving a mortality rate of 4.5. Gender stratification showed that 1,152 (7.4%) out of 15,622 males admitted died while 1,046 (3.1%) of the 33,665 of the females admitted died. Mortality was significantly higher in male gender than their female counterparts with odd ratio of 2.483 (2.278–2.704). Further stratification of the admitted patients based on route of admission showed that 37.0% of the patients were admitted via casualty (accident and emergency) while 63.0% were admitted via wards. A greater proportion of the males (59.8%) were admitted via casualty while the reverse was the case for the female patients (as they were admitted more via the ward; family medicine). The disparity was statistical significant (P< 0.05) (Table 1).

Table 2 shows the distribution of frequency of morbidity and mortality of the patients within the study period based on age. Age range 15–45 years had the highest number (57.9%) of admissions followed by 1–4 years (11.7%) and < 1 year (11.3%) age range. On the other hand, age range 15–45 years recorded the highest number (37.7%) of deaths followed by the 46–64 (23.8%) and the > 65 years (16.1%) category. However, the age range > 65 years

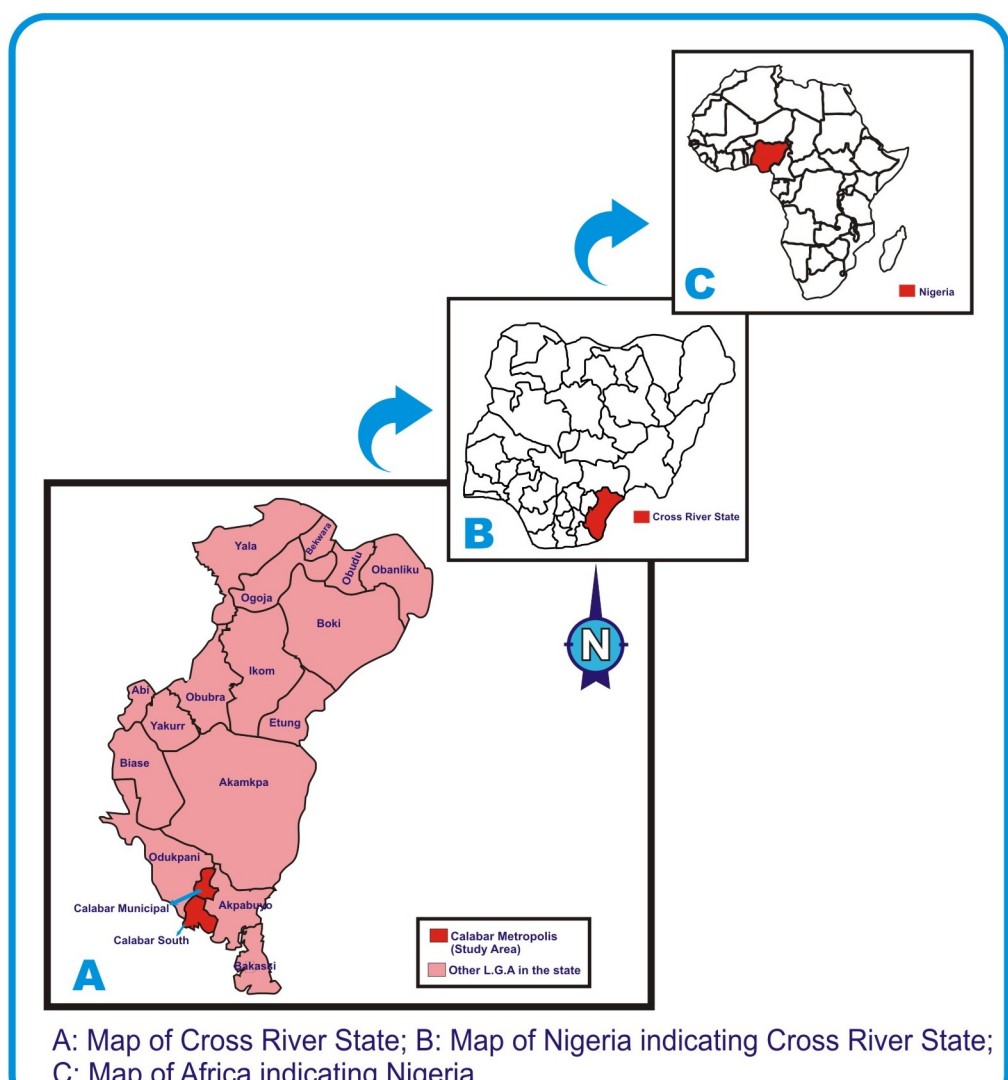

A: Map of Cross River State; B: Map of Nigeria indicating Cross River State; C: Map of Africa indicating Nigeria

**Fig 1. Map of the studied area.**

**Table 1. Frequency of morbidity and mortality of the studied population based on gender.**

| Gender | N. admitted (%) | Deaths (%) | M. rate (%) | $X^2$ | OR | P-value | CI |
|---|---|---|---|---|---|---|---|
| Male | 15,622 (31.7) | 1,152 (52.4) | 7.4 | 456.008 | 2.483 | <0.01 | 2.278–2.704 |
| Female | 33,665 (68.3) | 1,046 (47.6) | 3.1 | | 1 | | |
| Total | 49,287 (100.0) | 2,198 (100.0) | 4.5 | | | | |
| | Admission route | | | | | | |
| | Via ward (%) | Via casualty (%) (jjjjj(% % % | | 5108.04 | | <0.01 | |
| Male | 6,280 (40.2) | 9,342 (59.8) | | | | | |
| Female | 24,778 (73.6) | 8,887 (26.4) | | | | | |
| Total | 31,058 (63.0) | 18,229 (37.0) | | | | | |

N.: absolute number

M.: mortality

**Table 2. Frequency of morbidity and mortality of the studied population based on age.**

| Age (years) | Admissions (%) | Deaths (%) | M. rate (%) | $X^2$ | Df | P-value |
|---|---|---|---|---|---|---|
| <1 | 5,652 (11.5) | 253 (11.5) | 4.5 | 4887.235[a] | 5 | 0.000 |
| 1–4 | 5,764 (11.7) | 138 (6.3) | 2.4 | 2973.53[b] | 1 | 0.000 |
| 5–14 | 2,954 (6.0) | 102 (4.6) | 3.4 | | | |
| 15–45 | 28,535 (57.9) | 829 (37.7) | 2.9 | | | |
| 46–64 | 4,255 (8.6) | 523 (23.8) | 12.3 | | | |
| ≥65 | 2,127 (4.3) | 353 (16.1) | 16.6 | | | |

M.: mortality

[a]: Pearson Chi-square coefficient

[b]: Mantel Haenzel test for trend (linear by linear association)

recorded the highest number of deaths per admission giving rise to 16.5% mortality rate. The mortality rate increased as the age increased. The linear association was found to be significant (p<0.05) using Mantel Haenzel test for trend (Table 2).

Mortality rates for the years 2012, 2013, 2014, 2015, 2016 and 2017 were found to be 3.6%, 3.9%, 4.1%, 4.2%, 6.1% and 5.1% respectively (Fig 2).

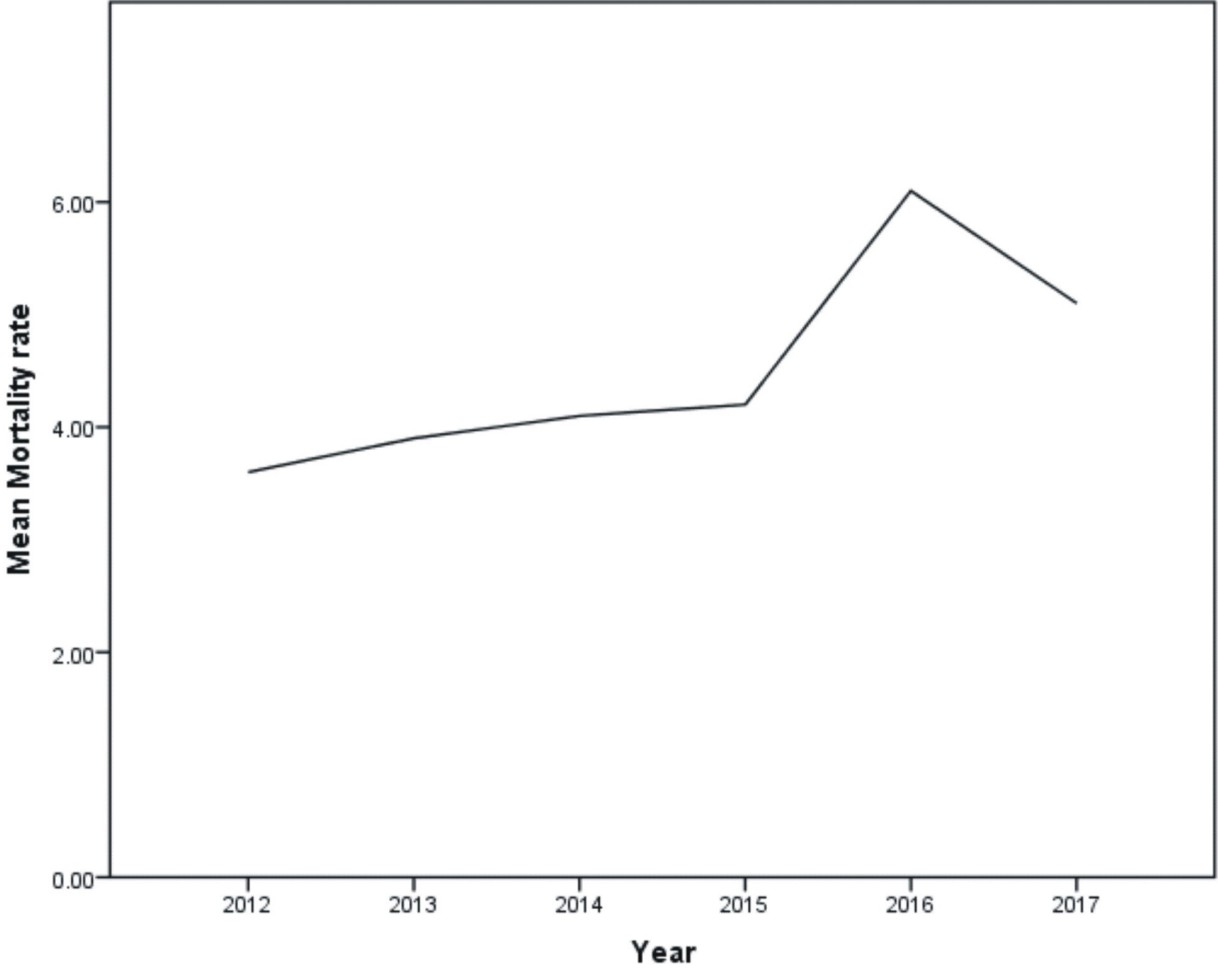

**Fig 2. Mortality rates of the studied years (2012–2017).**

**Table 3. Distribution of causes of morbidity and mortality by broad ICD-10 classification.**

| Disease diagnostic category (ICD-10 code) | Children (0–14 years) | | Adult (≥15 years) | | Total Adm. | Total death |
|---|---|---|---|---|---|---|
| | Admission | Death | Admission | Death | | |
| Infectious and parasitic diseases | 3,332 | 106 | 1,578 | 392 | 4,910 | 498 (22.7) |
| Neoplasms | 86 | 8 | 1,303 | 110 | 1,392 | 118 (5.4) |
| Disease of blood & blood forming organs | 549 | 6 | 415 | 61 | 964 | 67 (3.0) |
| Endocrine, nutritional & metabolic diseases | 859 | 24 | 857 | 90 | 1,716 | 114 (5.2) |
| Mental and behavioral disorders | 11 | 0 | 82 | 13 | 93 | 13 (0.6) |
| Diseases of the nervous system | 276 | 27 | 539 | 106 | 815 | 133 (6.0) |
| Eye and adnexa diseases | 162 | 0 | 300 | 0 | 462 | 0 (0.0) |
| Ear and mastoid process | 97 | 1 | 31 | 0 | 128 | 1 (0.0) |
| Circulatory system diseases | 239 | 27 | 2,301 | 321 | 2,540 | 348 (15.8) |
| Respiratory system diseases | 3,382 | 40 | 595 | 78 | 3,977 | 118 (5.4) |
| Diseases of the digestive system | 325 | 17 | 1,839 | 165 | 2,164 | 182 (8.3) |
| Diseases of the skin and subcutaneous tissue | 300 | 9 | 277 | 31 | 577 | 40 (1.8) |
| Diseases of musculoskeletal system & connective tissue | 114 | 10 | 282 | 13 | 396 | 23 (1.0) |
| Diseases of the genitourinary system | 299 | 17 | 1,397 | 105 | 1,626 | 122 (5.5) |
| Pregnancy, childbirth & puerperium | 10 | 1 | 16,108 | 18 | 16,118 | 19 (0.9) |
| Condition originating in the perinatal period | 2,497 | 123 | 1,752 | 2 | 4,249 | 125 (5.7) |
| Congenital malformations | 330 | 33 | 76 | 6 | 406 | 39 (1.8) |
| Symptoms, signs and abnormal clinical and laboratory findings | 636 | 15 | 639 | 88 | 1,275 | 103 (4.7) |
| Injury and poison & other external causes | 575 | 24 | 1,800 | 83 | 2,375 | 107 (4.9) |
| Factors influencing health status & contact with health services | 331 | 7 | 2,773 | 21 | 3,104 | 28 (1.3) |

Adm.: admission

Table 3 shows the causes of morbidity and mortality of the studied patient based on the broad ICD-10 classification. The broad ICD overall leading causes of death were infectious diseases and parasitic infections, and diseases of the circulatory system accounting for 22.7% and 15.8% of all deaths. However, diseases of the circulatory system recorded the highest number of deaths per admission (348/2,540) giving a mortality rate of 13.7%. Infectious diseases and parasitic infections were the leading causes of death in adults while conditions originating from perinatal period was the leading cause of death in children.

Table 4 shows further stratification of the specific causes of death within the broad ICD 10 classification based on adult and children classification.

On individual disease assessment of the causes of death, septicaemia (6.0%), stroke (4.2%), liver diseases (4.1%), tuberculosis (3.7%), diabetes (3.6%), complications of HIV/AIDS (3.5%) and ischaemic heart disease (3.4%) were the leading causes of mortality. Further stratification based on age showed malaria, diarrhea & gastroenteritis and acute pharyngitis and tonsillitis were the leading causes of admission while sepsis, chronic diseases of the tonsil and malaria were the leading causes of mortality in children. On the other hand, single spontaneous delivery, fetus affected by maternal factors & labor, post-partum hemorrhage (obstetric reasons) and diabetes mellitus (non-obstetric reason) were the leading causes of admission, while sepsis, stroke and liver diseases were the leading causes of death in adults (Table 5).

## Discussion

In sub-Saharan Africa, population-based information are scarce, hence, hospital-based morbidity and mortality data has become relevant surrogates in assessing disease burden, quality of health care as well as policy making [16].

**Table 4. Distribution of causes of morbidity and mortality by broad ICD-10 classification and specific disease types.**

| Diagnostic category (ICD-10 code) | Children (0–14 years) | | Adult (≥15 years) | | Total Adm. | Total death |
|---|---|---|---|---|---|---|
| | Admission | Death | Admission | Death | | |
| **Infectious and parasitic diseases** Infectious and parasitic diseases | | | | | | n = 498 (22.7) |
| Diarrhea & gastroenteritis | 1,154 | 17 | 131 | 14 | 1,285 | 31 (6.2) |
| Septicaemia | 327 | 38 | 269 | 94 | 596 | 132 (26.5) |
| HIV | 99 | 7 | 324 | 68 | 423 | 77 (15.5) |
| Other viral diseases | 46 | 1 | 244 | 75 | 290 | 76 (15.3) |
| Malaria | 1,450 | 23 | 145 | 10 | 1,595 | 33 (6.6) |
| Tuberculosis | 69 | 2 | 233 | 79 | 302 | 81 (16.3) |
| Others | 187 | 16 | 232 | 52 | 419 | 68 (13.6) |
| **Neoplasms** | | | | | | n = 118 (5.4) |
| Malignant neoplasm of lip, oral cavity | 1 | 0 | 19 | 0 | 20 | 0 (0.0) |
| Malignant neoplasm of liver | 1 | 0 | 58 | 20 | 59 | 20 (16.9) |
| Malignant neoplasm of breast | 3 | 0 | 141 | 20 | 144 | 20 (16.9) |
| Malignant neoplasm of cervix & uteri | 0 | 0 | 91 | 4 | 91 | 4 (3.4) |
| Benign neoplasm of breast | 1 | 0 | 19 | 0 | 20 | 0 (0.0) |
| Leiomyoma of uteri | 0 | 0 | 350 | 0 | 350 | 0 (0.0) |
| Others | 80 | 8 | 628 | 66 | 708 | 74 (62.7) |
| **Disease of blood & blood forming organs** | | | | | | n = 67 (3.0) |
| Anemia | 535 | 5 | 379 | 56 | 914 | 61 (91.) |
| Others | 14 | 1 | 36 | 5 | 50 | 6 (9.0) |
| **Endocrine, nutritional & metabolic diseases** | | | | | | n = 114 (5.2) |
| Diabetes mellitus | 15 | 0 | 664 | 78 | 679 | 78 (68.4) |
| Malnutrition | 135 | 13 | 9 | 1 | 144 | 14 (12.3) |
| Volume depletion | 656 | 6 | 16 | 0 | 672 | 6 (5.5) |
| Others | 53 | 5 | 168 | 11 | 221 | 16 (4.0) |
| **Mental and behavioral disorders** | | | | | | n = 13 (0.6) |
| **Diseases of the nervous system** | | | | | | n = 133 (6.0) |
| Inflammation disease of the CNS | 139 | 19 | 130 | 30 | 269 | 49 (36.8) |
| Cerebral palsy & other paralytic syndromes | 19 | 0 | 203 | 50 | 222 | 50 (37.6%) |
| Others | 118 | 8 | 206 | 26 | 324 | 34 (25.6) |
| **Diseases of the eye and adnexa diseases** | | | | | | n = 0 (0.0) |
| **Diseases of the ear and mastoid process** | | | | | | n = 1 (0.0) |
| **Circulatory system diseases** | | | | | | n = 348 (15.8) |
| Essential (primary) hypertension | 11 | 1 | 274 | 25 | 285 | 26 (7.5) |
| Other hypertensive diseases | 6 | 0 | 301 | 28 | 307 | 28 (8.0) |
| Ischaemic heart disease | 128 | 18 | 547 | 56 | 675 | 74 (21.3) |
| Stroke | 1 | 0 | 308 | 93 | 309 | 93 (26.7) |
| Others | 93 | 8 | 871 | 119 | 964 | 127 (36.5) |
| **Respiratory system diseases** | | | | | | n = 118 (5.4) |
| Acute pharyngitis and tonsillitis | 1018 | 1 | 30 | 0 | 1048 | 1 (0.8) |
| Other acute respiratory infections | 500 | 0 | 43 | 1 | 543 | 1 (0.8) |
| Pneumonia | 790 | 7 | 110 | 9 | 900 | 16 (13.6) |
| Acute bronchitis | 123 | 0 | 1 | 0 | 124 | 0 (0.0) |
| Chronic disease of the tonsils & adenoids | 388 | 26 | 21 | 0 | 409 | 26 (22.0) |
| Asthma | 372 | 1 | 51 | 0 | 323 | 1 (0.8) |
| Others | 291 | 5 | 339 | 68 | 630 | 73 (61.9) |

(*Continued*)

**Table 4.** (Continued)

| Diagnostic category (ICD-10 code) | Children (0–14 years) | | Adult (≥15 years) | | Total Adm. | Total death |
|---|---|---|---|---|---|---|
| | Admission | Death | Admission | Death | | |
| **Diseases of the digestive system** | | | | | | **n = 182 (8.3)** |
| Gastric and duodenal ulcer | 18 | 0 | 201 | 10 | 219 | 10 (5.5) |
| Diseases of the appendix | 45 | 0 | 336 | 2 | 381 | 2 (1.1) |
| Inguinal hernia | 63 | 0 | 274 | 11 | 337 | 11 (6.1) |
| Paralytic ileus & intestinal obstruction | 65 | 5 | 117 | 8 | 182 | 13 (7.1) |
| Alcohol liver disease | 0 | 0 | 11 | 4 | 11 | 4 (2.2) |
| Other diseases of the liver | 15 | 3 | 305 | 84 | 320 | 87 (7.8) |
| Others | 119 | 9 | 595 | 46 | 714 | 55 (30.2) |
| **Diseases of the skin & subcutaneous tissue** | | | | | | **n = 40 (1.8)** |
| **Diseases of musculoskeletal system & connective tissue** | | | | | | **n = 23 (1.0)** |
| **Diseases of the genitourinary system** | | | | | | **n = 122 (5.5)** |
| Acute nephritic syndrome | 42 | 10 | 241 | 40 | 283 | 50 (41.0) |
| Real tubule interstitial | 3 | 3 | 102 | 12 | 141 | 15 (12.3) |
| Other diseases of the urinary system | 99 | 4 | 349 | 46 | 448 | 50 (41.0) |
| Others | 53 | 0 | 705 | 7 | 758 | 7 (5.7) |
| **Pregnancy, childbirth & puerperium** | | | | | | **n = 19 (0.9)** |
| Medical abortion | 0 | 0 | 506 | 5 | 506 | 5 (26.3) |
| Post-partum hemorrhage | 2 | 0 | 759 | 1 | 761 | 1 (5.3) |
| Single spontaneous delivery | 3 | 0 | 5,717 | 1 | 5717 | 1 (5.3) |
| Others | 5 | 1 | 9,130 | 11 | 9,135 | 12 (63.1) |
| **Condition originating in the perinatal period** | | | | | | **n = 125 (5.7)** |
| Fetus affected by maternal factors & Labor | 25 | 0 | 1,005 | 0 | 1,030 | 0 |
| Slow fetal growth, malnutrition & short gest. | 330 | 21 | 82 | 0 | 412 | 16.8 |
| Congenital infectious & parasitic diseases | 867 | 23 | 231 | 1 | 1,098 | 19.2 |
| Others | 1,275 | 79 | 434 | 1 | 1,709 | 64.0 |
| **Congenital malformations** | | | | | | **n = 39 (1.8)** |
| **Symptoms, signs and abnormal clinical and laboratory findings** | | | | | | **n = 103 (4.7)** |
| **Injury and poison & other external causes** | | | | | | **n = 107 (4.9)** |
| Fracture of limb bone | 40 | 0 | 263 | 4 | 303 | 4 (3.7) |
| Burns and corrosion | 120 | 18 | 123 | 21 | 243 | 39 (36.5) |
| Others | 415 | 6 | 1,414 | 58 | 1,829 | 64 (59.8) |
| **Factors influencing health status & contact with health services** | | | | | | **n = 28 (1.3)** |

In this study, we observed mortality rate of 4.5%. This value is lower than 6.3% and 12.0% being studies in Ondo [13] and Kano [14], Nigeria. However, this observation is higher than an earlier hospital based study in Pakistan that reported mortality rate of 1.6% [24].

Gender stratification showed higher mortality in the male gender despite the higher admission rate of the females. This trend is similar to earlier reports from studies in Nigeria [4,14,] as well as another study in Ethiopia [25]. Generally, females have been shown to have lower mortality and relative longer life expectancy when compared to males [26–28]. This disparity has been attributed to higher mortality through injuries in males in Africa, Latin America, Caribbean and Europe [29]. However, a closer observation in the mode of admission offers a clue on the gender based hazard and treatment seeking behavior of the males in the studied area. Despite consisting 31.7% of the total admissions, males topped (59.8%) admission via accident and emergency/casualty ward. More males tends to be drivers and are more likely to

**Table 5. Summary of the leading seven specific causes of mortality in the study.**

| Specific disease | % of total deaths (%) | Rank | Specific disease | | | | | | | |
|---|---|---|---|---|---|---|---|---|---|---|
| | | | Children (0–14 years) | | | | Adult (≥15 years) | | | |
| | | | Admission (n = 14,370) | % | Deaths (n = 493) | % | Admission (n = 34,917) | % | Deaths (n = 1,705) | % |
| Sepsis | 6.0 | 1st | Malaria | 10.0 | Sepsis | 7.7 | Single spontaneous delivery | 16.4 | Sepsis | 5.5 |
| Stroke | 4.2 | 2nd | Diarrhea & gastroenteritis | 8.0 | Chronic disease of the tonsils & adenoids | 5.3 | Fetus affected by maternal factors & Labor | 2.9 | Stroke | 5.4 |
| Liver diseases | 4.1 | 3rd | Acute pharyngitis and tonsillitis | 7.1 | Malaria | 4.7 | Post-partum hemorrhage | 2.2 | Liver diseases | 4.9 |
| Tuberculosis | 3.7 | 4th | Congenital infection & parasitic diseases | 6.0 | Congenital infection & parasitic diseases | 4.7 | Diabetes | 1.9 | Tuberculosis | 4.6 |
| Diabetes | 3.6 | 5th | Pneumonia | 5.5 | Slow fetal growth, malnutrition & short gest. | 4.2 | Anemia | 1.1 | Diabetes | 4.4 |
| HIV | 3.5 | 6th | Volume depletion | 4.6 | Inflammation disease of the CNS | 3.8 | Leiomyoma of uteri | 1.0 | Other viral diseases | 4.4 |
| Other viral diseases | 3.5 | 7th | Anemia | 3.7 | Burns | 3.6 | Other diseases of the urinary system | 0.9 | HIV | 4.0 |
| Ischemic heart disease | 3.4 | 8th | Chronic disease of the tonsils & adenoids | 2.7 | Ischaemic heart disease | 3.6 | Diseases of the appendix | 1.0 | Ischemic heart disease | 3.3 |
| Acute nephritic syndrome | 2.3 | 9th | Asthma | 2.6 | Diarrhea & gastroenteritis | 3.4 | HIV | 0.9 | Anemia | 3.3 |
| Cerebral palsy & other paralytic syndromes | 2.3 | 10th | Slow fetal growth, malnutrition & short gest. | 2.3 | Malnutrition | 2.6 | Stroke | 0.8 | Cerebral palsy & other paralytic syndromes | 2.9 |
| Other diseases of urinary system | 2.3 | | | | | | | | | |

be victims of road accident, hence, emergency admission. More so, the cultural dogma of males being "bread winners" of their families cannot be ruled out as a contributing factor to deferred health seeking behaviors. However, the addition of admissions for labour and delivery which are not actually causes of morbidities may contribute to this disparity.

Majority (37.7%) of the death recorded in this study were observed in the 15–45 years age group. This pattern is similar to earlier reports from studies in other parts of Nigeria [4,16]. This observation is prevalent in most studies originating from Africa, mostly sub-Sahara Africa where life expectancy is short [4,16]. Life expectancy among Nigerians has shuttered between 57.2–65.9 years and 54.1–62.8 years for females and males, respectively [30].

Infectious and parasitic diseases were the leading causes of mortality in this study. The finding is in consonance with previous reports in developing countries [4,9,16,24,25]. This is also in agreement with the report of WHO 2004 Global Burden of disease for low income countries [25]. On further stratification, septicaemia, tuberculosis and complications of HIV were the chief causes of death due to infectious diseases accounting for 26.5, 16.3 and 15.5% of all deaths by infectious and parasitic disease, respectively. Sepsis (a consequence of septicaemia), a syndrome of deregulated host response to infection leading to life threatening organ dysfunction, is a major global health burden [31] causing about 5–6 million deaths annually with majority occurring in low and middle income countries [31,32]. Overall, septicaemia was the leading cause of both child and adult mortality and is responsible for 6.0% (132/2,198) of all deaths recorded in this study. Tuberculosis and complications of HIV accounted for 3.7 and 3.5% of the overall death becoming the 4th and 6th leading individual disease causes of death in this

study. Both have been reported as the leading cause of death in sub-Saharan Africa [1,33,34]. Despite the decline trend in tuberculosis globally, multidrug resistant tuberculosis (MDR-TB) has encouraged the epidemic in low-income countries with the incidence rate not less than 20 times higher in-low income countries than their high income counterparts [35–37]. Human immunodeficiency virus (HIV) on the other hand, has had its highest toll of epidemic in the sub-Saharan Africa with approximately 1 in every 25 adults living with HIV [38,39] and has been reported together with tuberculosis as leading cause of death in northwest Ethiopia [1]. Nigeria is now the second largest HIV disease burden in the world after South Africa which has 7.1 million (19% of global epidemic) burden of the disease, though prevalence is stable at 3.4% [38,39]. Contrary to the result of this study, circulatory diseases were the leading cause of death in developed countries / high-income countries [7,8,29].

Diseases of the circulatory system (cardiovascular and neurovascular diseases) were the second leading broad ICD-10 cause of death in this study. This finding is similar to that reported by Nwafor and colleagues in southeastern Nigeria [4]. Majority of the deaths due to circulatory diseases were caused by stroke (cerebrovascular accident) (26.7%) and ischaemic heart disease (21.3%). Aside being the chief cause of death due to circulatory diseases, stroke was observed to be the overall second leading individual disease cause of death in this study accounting for 4.2% of all deaths in the study (93/2198) above liver diseases, tuberculosis, diabetes and HIV/AIDS. Similarly, Arodigwe and colleagues have reported stroke as the second leading cause of mortality in southeastern Nigeria [16]. In sub-Saharan Africa, circulatory diseases' incidence has reached near epidemic proportion with preponderance of stroke, hypertension, cardiomyopathies and rheumatic heart disease reported as chief causes of mortality [40].

Diseases of the digestive system were the 3rd leading broad ICD-10 cause of death in this study. The major contributor to mortality observed in this category is liver diseases. Liver diseases represented the overall 3rd leading cause of death accounting for 4.1% of all deaths recorded in the study only behind sepsis and stroke. Liver diseases has not been implicated in previous studies as top cause of mortality. However, this emerging demographic calls for concern and might not be unconnected to rise in hepatitis B and C [39] which play major role in pathophysiology of most liver diseases. Unlike HIV/AIDS, treatment of hepatitis B and C in Nigeria is still via "out-of-pocket" of the patient. There is no programme for free treatment of both. The observation of this study calls for urgent intervention in this regard.

One of the components of the Sustainable development goals (Millennium Development Goals) is to reduce child mortality [41]. Conditions originating in the perinatal period constituted 5.7% of all causes of mortality, hence, ranking the overall 5th broad cause of mortality. Neonatal death is an important index used in evaluating socioeconomic development as well as an important indicator of status of a community [4,42]. Basically, it reflects the quality of prenatal, delivery and early infant care practices prevalent in any setting [4].

Neoplasms (together with diseases of the respiratory system) were the 7th broad leading cause of death accounting for 5.4% of the observed broad ICD-10 cause of death in this study. The specific diseases mainly involved in the mortality were malignant neoplasm of the breast and liver. This trend is consistent with previous finding in south eastern Nigeria [4]. However, infective agents such as hepatitis B and C are risk factors for the liver neoplasm [43].

Although the ICD-10 broad category of endocrine, nutritional and metabolic diseases ranked the 8th cause of death, diabetes mellitus as a single disease entity ranked the 4th overall cause of death in this study accounting for 3.5% of all deaths. Although Nigeria houses the highest number of persons living with diabetes in Africa [44], the mortality has not been recorded high in the past decades. However, a more recent (2016) WHO report documented diabetes mellitus as being responsible for 2% of all deaths in Nigeria [45]. The growing mortality due to diabetes as observed in this study is an indication of transition in disease burden.

This is possibly due to rapidly changing demographic trends, increasing rate of urbanization and transient adoption of western life style in many African settings [46,47].

Malaria, diarrhea & gastroenteritis, acute pharyngitis were the top causes of admission in children in this study. This finding is similar to previous studies in Nigeria that reported malaria, diarrhea/gastrointestinal diseases as the top causes of child morbidity in Bayelsa [48] and Delta [49] states, Nigeria. On the other hand, sepsis, chronic diseases of the tonsil and adenoids and malaria were the leading causes of death in children in this study. Similar to the finding of this study, Duru and colleagues have reported anemic heart failure (19.6%), sepsis (12.8%) and diarrhea (11.3%) as the leading causes of child death in Bayelsa State [48], while Muoneke and colleagues reported malaria (37.5%), gastroenteritis (23.6%) and broncho pneumonia (15.3%) as the leading causes of death in Ebonyi State [50]. Similarly, Ezeonwu *et al.*, have reported malaria (24.4%), sepsis (19.9%) and respiratory infections (7.7%) as the top causes of child mortality in Delta State [49]. Sepsis is life threatening organ dysfunction caused by deregulated host response to infection [51]. Bacteria (gram positive and gram negative) are the chief culprits of sepsis [52]. However, fungi causes do occur [53]. Chronic diseases of the tonsils and adenoids are inflammatory conditions resulting from proliferation and infection of adenoids and tonsils by bacterial agents. The chief bacterial culprits are *Haemophilus influenza*, *Streptococcus pneumonia*, *Staphylococcus aureus* and more [54]. On the other hand, malaria is caused by protozoa of the genus plasmodium specie and transmitted by female anopheles mosquito [55,56]. Diarrhea and gastroenteritis are mainly caused by rotavirus, norovirus, Salmonella, E. coli, campylobacter and others [57]. The above listed diseases are mostly preventable via simple and less cost effective measures. While sepsis and diarrhea can be ameliorated via application of simple hygiene practice, diseases of tonsils and adenoids can be reduced by childhood vaccination against some of the causative agents such as *Haemophilus influenza* and *Streptococcus spp* [48]. More so, malaria can be prevented by the use of insecticide treated bed nets and malaria chemoprophylaxis [48,58].

On the part of the adults, the leading causes of hospital admissions were single spontaneous delivery, fetus affected by maternal factors and post-partum hemorrhage (which are all obstetric issues) and diabetes. Although diabetes mellitus was the top non obstetric cause of admission, it did not contribute to the top causes of mortality in adults. On the other hand, sepsis, stroke and liver diseases were the leading causes of death in adults in this study. While sepsis is uncontrolled immunological response to infection, stroke is a cerebrovascular accident that leads to loss of brain function due to disruption of blood supply to the brain [59]. Sepsis is a major cause of death in children and adults. There were an estimated 10 million sepsis related deaths globally in 2017 with higher inclination in low- and middle-income countries [60]. Improved hand hygiene practice have been documented to reduce the incidence and consequent mortality due to sepsis [61]. On the other hand, stroke is mainly disease of the adult. Age is a strong determinant of stroke and the risk doubles every decade above age 55 [62].

It is pertinent to know that malaria is the top cause of morbidity and the third cause of mortality in children, whereas same did not apply to adults. In malaria endemic regions, several acquired and adaptive immunity have been documented [63,64]. These adaptive immunity are more developed and advanced in adults.

Amenable mortality implies deaths due to causes that otherwise shouldn't result to death in the presence of effective medical practice [2]. It is an indicator of national levels of personal health care access and quality [5]. The high level of mortality from infectious diseases and conditions originating from perinatal period reflects low access to quality health care in the studied population. This observation could be attributed to low contribution to health expenditure

by Nigerian government. The Financial Global Health database capped Nigeria health expenditure at $71 per person with 8.5% ($6) from development for assistance for health, 14.1% ($10) from government health spending, 76.1% ($54) from out-of-pocket spending and 1.4% ($1) from prepaid private spending [30].

Although infectious diseases constituted the majority of the causes of death observed in this study, diseases of the circulatory system recorded the highest mortality rate (13.7%) in relation to infectious diseases that had 10.1%. This is an indication for need to improve in research, practice, provision of facilities and policies in the area of circulatory / cardiovascular diseases.

## Limitations

The result of this study is potentially prone to varying limitations. Firstly, the study took a retrospective approach, hence, inherent limitations of retrospective studies such as selective bias might not be ruled out. More so, exact causes of death were based on clinical and ancillary investigations rather than postmortem examination (autopsy). Autopsy is not a common norm in the studied area due to some cultural dogmas except in cases of conflict or jurisprudence. Also, labour and delivery may have favored higher admission in females.

## Conclusion

The data in this study showed infectious disease and circulatory system diseases as the major causes of mortality in the studied population which reflects the common mortality pattern in developing countries. Aside sepsis, stroke was the second leading cause of mortality. The study revealed double burden of both communicable and non-communicable diseases. However, Infectious and parasitic diseases, Condition originating in the perinatal period, Respiratory system diseases were the leading causes of morbidity, with malaria being the chief individual cause of morbidity. Septicaemia, chronic disease of the tonsils and adenoids and malaria were the chief causes of mortality in children, while sepsis, stroke and liver diseases were the leading causes of death in adults. We thus recommend simultaneous intervention in circulatory diseases alongside with infectious diseases.

## Supporting information

**S1 Data.**
(ZIP)

## Acknowledgments

We appreciate Mr. Samuel Oscar, Mrs. Nwaiwu Patience Ndidi and Ms. Uchenwa Mercy for their meticulous role in data entering in this study.

## Author Contributions

**Conceptualization:** Henshaw Uchechi Okoroiwu.

**Data curation:** Henshaw Uchechi Okoroiwu, Kingsley Ikenna Uchendu, Rita A. Essien.

**Formal analysis:** Henshaw Uchechi Okoroiwu, Kingsley Ikenna Uchendu, Rita A. Essien.

**Methodology:** Henshaw Uchechi Okoroiwu.

**Resources:** Henshaw Uchechi Okoroiwu.

**Validation:** Kingsley Ikenna Uchendu, Rita A. Essien.

**Writing – original draft:** Henshaw Uchechi Okoroiwu.

**Writing – review & editing:** Kingsley Ikenna Uchendu, Rita A. Essien.

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
