## [Decision Letter · Decision Letter 0]

10 Dec 2019

PONE-D-19-27025

CAUSES OF MORBIDITY AND MORTALITY IN A TERTIARY HOSPITAL IN SOUTHERN NIGERIA; A 6 YEAR EVALUATION

PLOS ONE

Dear Mr Okoroiwu,

Thank you for submitting your manuscript to PLOS ONE. After careful consideration, we feel that it has merit but does not fully meet PLOS ONE’s publication criteria as it currently stands. Therefore, we invite you to submit a revised version of the manuscript that addresses the points raised during the review process.

We would appreciate receiving your revised manuscript by Jan 24 2020 11:59PM. To enhance the reproducibility of your results, we recommend that if applicable you deposit your laboratory protocols in protocols.io, where a protocol can be assigned its own identifier (DOI) such that it can be cited independently in the future. For instructions see: http://journals.plos.org/plosone/s/submission-guidelines#loc-laboratory-protocols

We look forward to receiving your revised manuscript.

Kind regards,

Chiara Lazzeri

Academic Editor

PLOS ONE

2. Please change your reference to "p=0.000" to "p<0.001" or as similarly appropriate, as p values cannot equal zero

4. Your ethics statement must appear in the Methods section of your manuscript. If your ethics statement is written in any section besides the Methods, please move it to the Methods section and delete it from any other section. Please also ensure that your ethics statement is included in your manuscript, as the ethics section of your online submission will not be published alongside your manuscript.

Reviewers' comments:

Reviewer's Responses to Questions

**Comments to the Author**

1. Is the manuscript technically sound, and do the data support the conclusions?

Reviewer #1: Partly

Reviewer #2: Partly

2. Has the statistical analysis been performed appropriately and rigorously? 

Reviewer #1: No

Reviewer #2: Yes

3. Have the authors made all data underlying the findings in their manuscript fully available?

Reviewer #1: Yes

Reviewer #2: Yes

4. Is the manuscript presented in an intelligible fashion and written in standard English?

Reviewer #1: Yes

Reviewer #2: No

5. Review Comments to the Author

Reviewer #1: The study is on the causes of morbidity and mortality in a tertiary hospital in Southern Nigeria and a retrospective work.

1. Being a retrospective work, were the authors able to verify accuracy of the diagnosis in each patient’s case notes to ascertain that the diagnosis was same as coded by the records? inputting wrong diagnosis is a common error in most hospitals in developing countries

2. The study was silent on missing data, a common problem with retrospective study.

3. To be more meaningful, the study should reclassify age groups of patients. The study lumped older children and young adult together which is not acceptable. Internationally, pediatric age group is from birth to <18 years; adults is 18 years to 64 years, and the elderly 65 years and above. This will allow for proper determination of leading cause of morbidity and mortality by age. Infectious diseases may not necessarily be the leading cause of morbidity and mortality among adults and elderly as concluded.

3. The study classified stroke as cardiovascular or circulation disease; Stroke is a cerebrovascular disorder and a neurological disorder and presently the leading cause of death in Nigeria like other developing countries. it is the commonest cause of disability. please reclassify stroke appropriately.

Reviewer #2: Given the paucity of data about in-hospital mortality and/or quality in many African countries, the paper does make a contribution to the literature. However, many points need to be clarified and/or expanded upon in order to increase the strength and interest of the article, as detailed below. The conclusions, particularly in the abstract, should also be more solidly explained as to their relation to the findings.

In addition, there are many mistakes in grammar which need to be corrected and I would recommend an editing service to review the manuscript.

Specific comments:

-Abstract conclusion: This does not give the reader a clear understanding of the main point/findings of the paper. It is very general. The meaning of "double burden of communicable and non-communicable diseases" needs to be specified. Also, if the authors wish to argue that most of the deaths in the study are preventable, this needs better supporting arguments in the discussion, as below.

-Methods: is this in-hospital mortality? If not, what duration of follow-up was used?

-Methods: should state whether this includes Labor and Delivery (see below corollary comment re: Discussion section)

-Methods: who determined the cause of death which is coded by ICD-10? The treating physician?

-Results: In the last line of the first paragraph, you stated that "mortality was significantly higher in female gender," however, the data suggest the opposite, and in the discussion section higher male mortality is discussed

-Results: Second paragraph, "647.0% of the patients were admitted via casualty"--the number needs to be correct to 0-100%

-Results: 5th paragraph: the text would be more informative/interesting with a description of what is meant specifically by "conditions originating from the perinatal period"

-Results: Description of which bacterial organisms the most common cause of septicemia would very much improve the strength

-Results: The last paragraph describes the percentage of deaths due to the leading causes listed.

(a) What does a death due to HIV/AIDS mean? Did these patients die from the virus itself (eg, wasting syndrome, etc.) or more likely, due to opportunistic infections? Again would be helpful to outline this more specifically, and if opportunistic infections, which ones?

(b) It would add significant strength and interest to the paper to know: Of the patients admitted with each of these conditions, what proportion died?

-Discussion: In the discussion of higher mortality among males, it would be important to know whether the female admissions/deaths include those admitted for labor and delivery, which would be expected to make the female mortality rate appear lower--eg, most would not be coming in "sick." If this did include L+D, would consider re-analyzing without those admissions to see if the difference in male and female mortality remains.

-Discussion: The commentary on males having less "health conscious" behavior than females seems somewhat like conjecture and should either be removed or re-stated using less definitive language (unless able to support with literature/data)

-Discussion: A rise in Hepatitis B and C is described. Would expand upon this more given that the finding of liver disease as one of the leading causes of death was the most surprising and potentially interesting piece of new information from the study. Can you specify rates of hepatitis viruses among those with liver disease who died in your study? If not, would describe if there are limitations in availability of testing. It would also be worthwhile to discuss how patients with these viral hepatitides are managed what at your facility and/or in Nigeria in general--are treatments available? Is this data suggesting a reason call for more specific resources for HBV and HCV?

-Discussion: the paragraph regarding conditions in the perinatal period and the Millenium Development Goals does not seem well-placed, and it is not clear how what is written here adds to the literature

-Discussion: Breast and liver cancer were described as leading causes of neoplasms, which is stated is consistent with previous findings. Previous findings from where? Similar hospital settings/LMIC's, etc? Should specify. Also, liver cancer as a leading neoplasm provides yet another opportunity to discuss more specifics about viral hepatitis, and connect to the above comments.

-Discussion: The argument that the mortality from infections and "conditions originating in the perinatal period" reflects low access to quality health care and reflects "amenable mortality" needs to be better supported and explained. As currently written, this reads as an assumption. Ideas include concepts such as: Preventable at what stage? If a patient presents with late-stage septic shock, just because there was an infectious cause does not necessarily mean this death could be prevented even with maximal quality of care at hospital presentation, etc.

-Discussion: I found the discussion about malaria as a cause of morbidity distracting, since the paper is focused on mortality

6. PLOS authors have the option to publish the peer review history of their article (what does this mean?). If published, this will include your full peer review and any attached files.

Reviewer #1: Yes: Prof E.O. Sanya

Reviewer #2: No

---

## [Author Response · Author response to Decision Letter 0]

24 Jul 2020

Response to review comments

Reviewer #1: The study is on the causes of morbidity and mortality in a tertiary hospital in Southern Nigeria and a retrospective work.

1. Being a retrospective work, were the authors able to verify accuracy of the diagnosis in each patient’s case notes to ascertain that the diagnosis was same as coded by the records? inputting wrong diagnosis is a common error in most hospitals in developing countries

Response:

They authors did not verify via case notes. The trained staff at health records departments do record same from case notes and submit same quarterly to Medical Advisory Committee.

2. The study was silent on missing data, a common problem with retrospective study.

Response:

There was none in the report retrieved based on information used.

3. To be more meaningful, the study should reclassify age groups of patients. The study lumped older children and young adult together which is not acceptable. Internationally, pediatric age group is from birth to <18 years; adults is 18 years to 64 years, and the elderly 65 years and above. This will allow for proper determination of leading cause of morbidity and mortality by age. Infectious diseases may not necessarily be the leading cause of morbidity and mortality among adults and elderly as concluded.

Response:

There was a mistake in the labelling earlier on children. Same has been corrected children age properly listed.

3. The study classified stroke as cardiovascular or circulation disease; Stroke is a cerebrovascular disorder and a neurological disorder and presently the leading cause of death in Nigeria like other developing countries. it is the commonest cause of disability. please reclassify stroke appropriately.

Response:

The suggestion has been done.

Reviewer #2: Given the paucity of data about in-hospital mortality and/or quality in many African countries, the paper does make a contribution to the literature. However, many points need to be clarified and/or expanded upon in order to increase the strength and interest of the article, as detailed below. The conclusions, particularly in the abstract, should also be more solidly explained as to their relation to the findings.

In addition, there are many mistakes in grammar which need to be corrected and I would recommend an editing service to review the manuscript.

Response:

We have put more effort to rid off typological errors and mistakes.

Specific comments:

-Abstract conclusion: This does not give the reader a clear understanding of the main point/findings of the paper. It is very general. The meaning of "double burden of communicable and non-communicable diseases" needs to be specified. Also, if the authors wish to argue that most of the deaths in the study are preventable, this needs better supporting arguments in the discussion, as below.

Response:

We have added explanation to the earlier statement.

-Methods: is this in-hospital mortality? If not, what duration of follow-up was used?

Response:

It is an in-hospital mortality. Further details has been provided in the methods section.

-Methods: should state whether this includes Labor and Delivery (see below corollary comment re: Discussion section)

Response:

This information has been added.

-Methods: who determined the cause of death which is coded by ICD-10? The treating physician?

Response:

The attending physicians did. This information has been added to the method section.

-Results: In the last line of the first paragraph, you stated that "mortality was significantly higher in female gender," however, the data suggest the opposite, and in the discussion section higher male mortality is discussed

Response:

The error has been corrected.

-Results: Second paragraph, "647.0% of the patients were admitted via casualty"--the number needs to be correct to 0-100%

Response:

The error has been corrected.

-Results: 5th paragraph: the text would be more informative/interesting with a description of what is meant specifically by "conditions originating from the perinatal period"

Response:

This is the terminology for the ICD-10 classification of the group. Details has been done in the new part of the discussion added as the mortality discussion was done for children and adults separately.

-Results: Description of which bacterial organisms the most common cause of septicemia would very much improve the strength

Response:

This information has been added in the discussion.

-Results: The last paragraph describes the percentage of deaths due to the leading causes listed.

(a) What does a death due to HIV/AIDS mean? Did these patients die from the virus itself (eg, wasting syndrome, etc.) or more likely, due to opportunistic infections? Again would be helpful to outline this more specifically, and if opportunistic infections, which ones?

(b) It would add significant strength and interest to the paper to know: Of the patients admitted with each of these conditions, what proportion died?

Response:

We have used complications of HIV/AIDS. We do not have data on specific complications and how many died for each.

-Discussion: In the discussion of higher mortality among males, it would be important to know whether the female admissions/deaths include those admitted for labor and delivery, which would be expected to make the female mortality rate appear lower--eg, most would not be coming in "sick." If this did include L+D, would consider re-analyzing without those admissions to see if the difference in male and female mortality remains.

Response:

We have stated this in the methods and as well as in the discussion. We were not able to re stratify data deleting obstetric cases. Entered as part of the limitations.

-Discussion: The commentary on males having less "health conscious" behavior than females seems somewhat like conjecture and should either be removed or re-stated using less definitive language (unless able to support with literature/data)

Response:

This has been deleted.

-Discussion: A rise in Hepatitis B and C is described. Would expand upon this more given that the finding of liver disease as one of the leading causes of death was the most surprising and potentially interesting piece of new information from the study. Can you specify rates of hepatitis viruses among those with liver disease who died in your study? If not, would describe if there are limitations in availability of testing. It would also be worthwhile to discuss how patients with these viral hepatitides are managed what at your facility and/or in Nigeria in general--are treatments available? Is this data suggesting a reason call for more specific resources for HBV and HCV?

Response:

The suggestion has been inputed. However, we do not have information on rate of hepatitis virus in the subjects with liver diseases.

-Discussion: the paragraph regarding conditions in the perinatal period and the Millenium Development Goals does not seem well-placed, and it is not clear how what is written here adds to the literature

Response:

We have placed it properly.

-Discussion: Breast and liver cancer were described as leading causes of neoplasms, which is stated is consistent with previous findings. Previous findings from where? Similar hospital settings/LMIC's, etc? Should specify. Also, liver cancer as a leading neoplasm provides yet another opportunity to discuss more specifics about viral hepatitis, and connect to the above comments.

Response:

The location of the finding has been inserted. 

-Discussion: The argument that the mortality from infections and "conditions originating in the perinatal period" reflects low access to quality health care and reflects "amenable mortality" needs to be better supported and explained. As currently written, this reads as an assumption. Ideas include concepts such as: Preventable at what stage? If a patient presents with late-stage septic shock, just because there was an infectious cause does not necessarily mean this death could be prevented even with maximal quality of care at hospital presentation, etc.

Response:

More clarification has been done via the new part of discussion in the preventive means to avoid some of the mortalities.

-Discussion: I found the discussion about malaria as a cause of morbidity distracting, since the paper is focused on mortality

Response:

The problem has been rectified by simultaneous discussion of mortality in children and adult in the new part of the discussion introduced. 

All corrections, inputs, adjustments and insertions are highlighted in red. Only deletions are not visible.

---

## [Editor Report · Decision Letter 1]

27 Jul 2020

CAUSES OF MORBIDITY AND MORTALITY AMONG PATIENTS ADMITTED IN A TERTIARY HOSPITAL IN SOUTHERN NIGERIA; A 6 YEAR EVALUATION

PONE-D-19-27025R1

Dear Dr. Okoroiwu,

We’re pleased to inform you that your manuscript has been judged scientifically suitable for publication and will be formally accepted for publication once it meets all outstanding technical requirements.

Kind regards,

Chiara Lazzeri

Academic Editor

PLOS ONE
---

## [Editor Report · Acceptance letter]

3 Aug 2020

PONE-D-19-27025R1 

CAUSES OF MORBIDITY AND MORTALITY AMONG PATIENTS ADMITTED IN A TERTIARY HOSPITAL IN SOUTHERN NIGERIA; A 6 YEAR EVALUATION 

Dear Dr. Okoroiwu:

I'm pleased to inform you that your manuscript has been deemed suitable for publication in PLOS ONE. Congratulations! Your manuscript is now with our production department. 

Kind regards, 

on behalf of

Dr. Chiara Lazzeri 

Academic Editor

PLOS ONE